# Holomics and Artificial Intelligence-Driven Precision Oncology for Medullary Thyroid Carcinoma: Addressing Challenges of a Rare and Aggressive Disease

**DOI:** 10.3390/cancers16203469

**Published:** 2024-10-13

**Authors:** Thifhelimbilu Emmanuel Luvhengo, Maeyane Stephens Moeng, Nosisa Thabile Sishuba, Malose Makgoka, Lusanda Jonas, Tshilidzi Godfrey Mamathuntsha, Thandanani Mbambo, Shingirai Brenda Kagodora, Zodwa Dlamini

**Affiliations:** 1Department of Surgery, University of the Witwatersrand, Johannesburg 2193, South Africa; maeyane.moeng@wits.ac.za (M.S.M.); nosisa.sishuba@yahoo.com (N.T.S.); 2Department of Surgery, University of Pretoria, Pretoria 0002, South Africa; malose.makgoka@up.ac.za; 3Department of Surgery, University of Limpopo, Mankweng 4062, South Africa; lusandajonas@yahoo.com (L.J.); tshilidzimamathuntsha@gmail.com (T.G.M.); 4Department of Surgery, University of KwaZulu-Natal, Durban 2025, South Africa; mbambot@gmail.com; 5Department of Nuclear Medicine, University of the Witwatersrand, Johannesburg 2193, South Africa; brenda.kagodora@wits.ac.za; 6SAMRC Precision Oncology Research Unit (PORU), DSI/NRF SARChI, Precision Oncology and Cancer Prevention (POCP), University of Pretoria, Pretoria 0028, South Africa; zodwa.dlamini@up.ac.za

**Keywords:** artificial intelligence, holomics, medullary thyroid carcinoma, precision oncology

## Abstract

**Simple Summary:**

Medullary thyroid carcinoma (MTC) is a rare but aggressive form of thyroid cancer accounting for over 10% of deaths related to thyroid malignancies. MTC can be either hereditary or sporadic. Although it can be cured if it is detected early and completely removed, most patients with MTC are diagnosed when the cancer has already spread beyond the thyroid gland, making it difficult to treat. Managing MTC is complex, despite the availability of newer treatment options like targeted therapy, which works in less than 30% of cases and can lead to severe side effects in some patients. MTC behaves differently in each patient, making the selection of appropriate treatments challenging, even for multidisciplinary teams of experts. This article aims to describe the challenges faced during the diagnostic workup and management of MTC patients. It highlights how holomics, which is an integrated approach combining various biological data types, and artificial intelligence (AI) can assist in improving patient outcomes. By simultaneously analyzing and integrating findings from biochemical, radiological, and histological investigations, genetic studies, and other sources, along with the personal information of a patient, AI can enhance decision-making processes. This innovative approach has the potential to personalize and optimize treatment strategies, leading to better management and improved outcomes for patients with MTC.

**Abstract:**

**Background/Objective:** Medullary thyroid carcinoma (MTC) is a rare yet aggressive form of thyroid cancer comprising a disproportionate share of thyroid cancer-related mortalities, despite its low prevalence. MTC differs from other differentiated thyroid malignancies due to its heterogeneous nature, presenting complexities in both hereditary and sporadic cases. Traditional management guidelines, which are designed primarily for papillary thyroid carcinoma (PTC), fall short in providing the individualized care required for patients with MTC. In recent years, the sheer volume of data generated from clinical evaluations, radiological imaging, pathological assessments, genetic mutations, and immunological profiles has made it humanly impossible for clinicians to simultaneously analyze and integrate these diverse data streams effectively. This data deluge necessitates the adoption of advanced technologies to assist in decision-making processes. Holomics, which is an integrated approach that combines various omics technologies, along with artificial intelligence (AI), emerges as a powerful solution to address these challenges. **Methods:** This article reviews how AI-driven precision oncology can enhance the diagnostic workup, staging, risk stratification, management, and follow-up care of patients with MTC by processing vast amounts of complex data quickly and accurately. Articles published in English language and indexed in Pubmed were searched. **Results:** AI algorithms can identify patterns and correlations that may not be apparent to human clinicians, thereby improving the precision of personalized treatment plans. Moreover, the implementation of AI in the management of MTC enables the collation and synthesis of clinical experiences from across the globe, facilitating a more comprehensive understanding of the disease and its treatment outcomes. **Conclusions:** The integration of holomics and AI in the management of patients with MTC represents a significant advancement in precision oncology. This innovative approach not only addresses the complexities of a rare and aggressive disease but also paves the way for global collaboration and equitable healthcare solutions, ultimately transforming the landscape of treatment and care of patients with MTC. By leveraging AI and holomics, we can strive toward making personalized healthcare accessible to every individual, regardless of their economic status, thereby improving overall survival rates and quality of life for MTC patients worldwide. This global approach aligns with the United Nations Sustainable Development Goal 3, which aims to ensure healthy lives and promote well-being at all ages.

## 1. Introduction

Medullary thyroid carcinoma (MTC) constitutes less than 5% of primary malignant tumors of the thyroid gland but more than 10% of the deaths from thyroid cancer [1]. Approximately 25% of MTC is hereditary and part of either multiple endocrine neoplasia type 2A or 2B (MEN 2A or MEN 2B) [2,3]. Eighty percent of hereditary MTC occurs in individuals with the classical variant of MEN 2A, while 5% occurs in those with MEN 2B. Variants of MTC in MEN 2A include the classical type in 80% and Hirschsprung’s disease and cutaneous lichen amyloidosis-associated disease in 1% each. Close to 15% of hereditable MTC is part of the familial medullary thyroid carcinoma (FMTC) variant of MEN 2A [3,4,5]. Around 53% and 20% of patients with sporadic MTC carry mutually exclusive somatic mutations of the rearranged during transfection (*RET*) and rat sarcoma virus (*RAS*) genes, respectively.

Aberrant splicing is a crucial factor in the development and progression of cancers, including MTC. Splicing removes introns from pre-mRNA and joins exons to form mature mRNA, which is a process regulated by the spliceosome and various splicing factors. Disruption of this regulation can result in abnormal mRNA isoforms that contribute to cancer [6]. The spliceosome consists of small nuclear RNAs (snRNAs) and associated proteins, with key splicing factors including serine/arginine-rich (SR) proteins and heterogeneous nuclear ribonucleoproteins (hnRNPs). The SR proteins usually promote exon inclusion, while hnRNPs can repress splicing or enhance exon skipping [7,8].

Medullary thyroid carcinoma is often associated with mutations in the *RET proto-oncogene*, which encodes a receptor tyrosine kinase involved in cell growth and differentiation. In addition to point mutations, alternative splicing of *RET* can produce different isoforms with varying oncogenic potentials. For example, the inclusion or exclusion of specific exons in the *RET* transcript can influence kinase activity, cellular localization, and interaction with signaling partners [9]. In MTC, aberrant splicing of the *RET* gene has been observed, leading to the expression of *RET* isoforms that contribute to the malignancy. Studies have identified splicing variants of *RET* that lack the entire exon 11 or 12, resulting in constitutively active forms of the receptor that promote uncontrolled cell growth [10]. Additionally, the aberrant splicing of other genes involved in cell cycle regulation, apoptosis, and metastasis has been implicated in MTC pathogenesis [11]. Targeting aberrant splicing represents a promising therapeutic strategy for MTC. Several approaches are being explored, including small molecules that modulate splicing factor activity and antisense oligonucleotides (ASOs) that correct splicing defects. For instance, spliceosome inhibitors like spliceostatin A and E7107 have shown potential in preclinical studies by inducing apoptosis in cancer cells with splicing abnormalities [12]. ASOs can be designed to bind to specific RNA sequences and modify splicing patterns, potentially restoring normal gene expression [13].

Recent advances have also explored the use of CRISPR/Cas9 technology to target and correct splicing mutations at the genomic level, offering another potential avenue for therapeutic intervention [14]. Aberrant splicing plays a significant role in the pathogenesis of MTC, particularly through the dysregulation of the *RET* proto-oncogene. Understanding the mechanisms underlying splicing abnormalities in MTC can reveal novel therapeutic targets and strategies, offering hope for more effective treatments for this aggressive cancer.

A diagnostic workup of patients with suspected MTC starts with a clinical assessment followed by a thyroid function test, ultrasound, measurement of serum level of Ctn, fine-needle aspiration cytology (FNAC), CT scan, and MRI [1]. The choice and timing of treatment and follow-up plans depend on the type of MTC, stage and grade of a tumor, mutational status, and serum levels of Ctn and CEA and their doubling time [1,3,15]. Chemotherapy, radiotherapy radioactive iodine, and other radionuclide therapies are not effective in the treatment of MTC [1,16]. Targeted therapy using multi- or selective tyrosine kinase inhibitors is effective in a few patients with MTC.

The management of patients with MTC is often not individualized, and a one-size-fits-all approach is followed, although MTC is known to be a highly complex and heterogeneous disease. The American Joint Committee on Cancer tumor node metastasis (TNM) staging system cannot accurately quantify the volume of the disease to enable personalized management of patients with MTC [17].

The level of serum Ctn is generally useful for screening, risk stratification, and postoperative monitoring following curative resection of MTC [18,19,20,21]. Other parameters for the prognostication of MTC include the age and gender of the patient, the size of the tumor, the existence of extra-thyroidal spread, lymph node metastasis, and levels of serum Ctn and carcinoembryonic antigen (CEA) [1,22]. The mutational landscape of a tumor, alterations in the expression of non-coding RNAs, and the molecular profile of the tumor may influence the clinical behavior and outcome of patients with MTC [23,24,25,26,27,28,29,30,31,32].

Medullary thyroid carcinoma is curable if surgical treatment is instituted when it is still confined to the thyroid gland and is completely removed [21,33,34,35,36,37]. A patient with MTC is considered cured if there is no structural evidence of residual disease and serum levels of Ctn and CEA are either normal or not detectable. However, the majority of patients with MTC present when the tumor is larger than 2 cm and there are metastases to lymph nodes, lungs, bones, or liver [21,38,39,40,41,42,43,44,45]. The prognosis in the majority of patients with MTC remains poor despite the introduction of newer diagnostic modalities and targeted therapy and the involvement of multidisciplinary teams in decision-making and management.

## 2. Management of Medullary Thyroid Carcinoma

The mainstay of the treatment of MTC is total thyroidectomy with central lymph node dissection [36,38]. Radioactive iodine, thyroid stimulating hormone (TSH) suppression, external beam radiotherapy, standard chemotherapy, and immunotherapy are not effective in the treatment of MTC [16,36,38,46]. Medullary thyroid carcinoma, especially if it is larger than 1 cm in size, may overexpress programmed cell death-ligand 1, which may make it amenable to treatment with PD-L1 inhibitors [47]. External beam radiotherapy is used to treat residual or recurrent disease in the neck that is not amenable to resection [1,16]. The tumor microenvironment (TME) of MTC, like most thyroid cancers, is less immunogenic; thus, immunotherapy is usually not effective [44]. Like targeted therapy with selective tyrosine kinase (TKIs and multi-kinase inhibitors (MKIs)), immunotherapy may slow down the progression of MTC and palliate symptoms like diarrhea in the case of metastatic MTC [48,49]. Occasionally, TKIs, when used as neo-adjuvant therapy, may lead to a significant reduction in the size of the tumor in patients with locally advanced and irresectable MTCs and make it amenable to a resection [50].

### 2.1. Extent of Thyroidectomy for Medullary Thyroid Carcinoma

Total thyroidectomy is indicated in all patients with hereditary MTC, as the tumor is likely to be multifocal or bilateral [51,52]. A thyroid lobectomy with prophylactic central lymph node dissection is sometimes appropriate in patients with a low-risk MTC. A low-risk MTC must be sporadic, unifocal, low-grade, and less than 1 cm in maximum diameter. To be low-risk, an MTC must also be intra-thyroidal and not harbor a high-risk mutation [36,53]. Additionally, the serum levels of both Ctn and CEA must normalize within 1–3 months after the lobectomy [54]. Among the concerns regarding thyroid lobectomy for the curative treatment of MTC include the possibility of mistaking late-onset hereditary MTC with the sporadic type. Furthermore, an MTC that is less than 2 cm may already have metastasized to regional lymph nodes or distant sites [53]. Furthermore, the serum level of Ctn is not always useful, as some patients with MTC may have normal levels or levels that are discordant with CEA results [5].

### 2.2. Extent of Cervical Lymphadenectomy

Medullary thyroid carcinoma should preferably be diagnosed preoperatively, and the patient should be offered total thyroidectomy and appropriate lymph node dissection as completion or re-do surgery, which is associated with a higher rate of postoperative complications and poorer outcomes [55]. If MTC was missed during the preoperative investigation, and the diagnosis is made following a diagnostic lobectomy, subsequent management may be observation with serial measurements of serum Ctn and CEA or completion thyroidectomy [33,53,54,56]. Although most experts would recommend a completion thyroidectomy, some do not deem it always necessary, as the incidence of bilateral disease in sporadic MTC is less than 10% [53]. The need for a completion thyroidectomy is however not debatable if the tumor has high-risk features like lymph node metastases, a markedly elevated serum level of Ctn, or mutations associated with aggressive disease [53]. Other markers of an aggressive MTC that would strengthen the need for a completion thyroidectomy include patient factors like male gender and older age and high-grade tumor on histology [4]. High-risk histological features include evidence of necrosis in the tumor, mitotic count above 4 per 2 mm^2^, and a Ki67 index greater than 4% [56,57]. Most experts, however, insist that a total thyroidectomy with central lymph node dissection should be the minimum treatment in any patient with MTC, regardless of the level of serum Ctn and the stage, grade, or mutational status of a tumor [36,51].

Patients with an MTC that is less than 4 cm in diameter and no clinical and/or radiological evidence of cervical lymph node involvement should have a total thyroidectomy and prophylactic central lymph node neck dissection [36,52,58]. Central lymph node dissection implies the removal of lymph nodes bearing tissues from the hyoid bone to the brachiocephalic vein inferiorly and laterally bordered by the right and left medial aspects of the internal jugular veins [51]. The clearance of central lymph nodes, even in patients with no evidence of metastatic involvement on clinical examination, in preoperative imaging investigations, and during surgery, increases the likelihood of cure and reduces the risk of local recurrence and the need for re-do surgery [51,59,60,61]. Some patients with MTC have metastasis to cervical lymph nodes that are missed during the clinical examination and imaging with investigations [62,63].

Although most surgeons routinely include central lymph node dissection during total thyroidectomy for MTC, some omit it if serum Ctn is less than 20 pg/mL in patients whose tumor is smaller than 1 cm in diameter and there is no evidence of lymph node involvement, as the likelihood of occult metastasis is low [39,64]. There are surgeons who would add a prophylactic central and ipsilateral lateral neck dissection if the preoperative Ctn level is above 20 pg/mL [20]. Additionally, prophylactic contralateral lateral cervical lymph node dissection is advised in patients with MTC and preoperative serum Ctn between 200 pg/mL and 1000 pg/mL, as the likelihood of achieving a biochemical cure in this group of patients is around 50% [49,50,51]. Biochemical cure is a postoperative level of serum Ctn that is not detectable or normal at least 1–3 months after surgery [65]. A normalized or undetectable level of Ctn without structural evidence of disease following surgery in patients with MTC is associated with 98% 10-year survival and a recurrence rate below 5% [66].

### 2.3. Management of Locally Advanced or Metastatic Medullary Thyroid Carcinoma

Treatment goals in patients with irresectable MTC are locoregional control and palliation of symptoms resulting from excess hormones like Ctn, pain management, and prevention or relief of airway compression [35]. Extensive surgery is generally discouraged in patients with MTC with extensive extra-thyroidal extension. The palliative resection of MTC may be necessary if the tumor is locally advanced but there is impending life-threatening involvement of the upper aerodigestive tract [35,67]. Among the surgical options in patients with irresectable MTC are debulking, laryngectomy, esophagectomy, laryngo-esophagectomy, and metastectomy [35,36]. Although there are isolated reports of the successful use of TKIs as neo-adjuvant therapy, there is generally no role for neo-adjuvant therapy in the management of locally advanced and irresectable MTC [50]. Metastectomy for palliation is justifiable in patients with MTC and isolated symptomatic metastatic lesions that are progressively enlarging [51,52]

Targeted therapy with *RET* or *VEGFR-2* inhibitors is currently the first-line therapy for the management of patients with metastatic MTC [35,36,52,68]. Cabozantinib and vandetanib were the first drugs to be approved by the Food and Drug Administration (FDA) as first-line TKIs for managing locally advanced or metastatic MTC [67]. Both cabozantinib and vandetanib can improve progression-free survival in patients with locally advanced or metastatic MTC [67]. The efficacy of XL184 (Cabozantinib) in the Advanced Medullary Thyroid Cancer (EXAM) trial showed that it improved median progression-free survival of around four months in the placebo to 11.2 months in the treatment group (95% CI 0.19–0.40) [67]. Due to the high risk of adverse events and toxicity of TKIs and MKIs, patient selection is crucial. Treatment with TKIs or MKIs should be limited to patients with high tumor burden with symptomatic disease or rapidly progressing disease and favorable proteo-metabolic status [67,68,69].

Other options for the management of locally advanced or metastatic MTC include peptide receptor radionuclide therapy [56], external beam radiotherapy [57], radiofrequency ablation, cryoablation, and embolization [35,36,51,52,68]. Medullary thyroid carcinoma is not sensitive to standard chemotherapeutic agents [51]. Similarly, external beam radiotherapy does not improve survival in patients with irresectable or residual tumors in patients with locally advanced MTC with extra-thyroidal extension [16]. The assessment of the stability of advanced MTC uses either the trend of biochemical markers like serum Ctn and/or the response evaluation criteria in solid tumors (RECIST) criteria [35,51,52,67,68].

### 2.4. Postoperative Follow-Up

Patients with MTC should be assessed clinically every 3–6 months following surgery [1]. Serial serum Ctn and CEA measurement is performed every 6 months for two years and yearly subsequently; if there is no evidence of recurrent disease, an ultrasound of the neck should be performed every 3 to 12 months. Persistently elevated postoperative serum Ctn levels indicate residual disease, and the magnitude of the elevation influences further evaluation and management [44]. If after 2 to 3 months following surgery, the level of Ctn is above 150 pg/mL, there is a high likelihood that the patient has metastatic disease, which necessitates a metastatic workup that should include a CT scan of the neck and abdomen [36,70,71]. Metastases from MTC are usually small and may be missed by a CT scan. Other imaging investigations like a bone scan, MRI, and 18F-FDG-PET/CT should be added if metastases are suspected but not visible on ultrasound and CT scan [21,36,51,72,73,74]. Nevertheless, 18F-FDG-PET/CT is recommended in patients with MTC and serum levels of Ctn of at least 500 to 1000 pg/mL with no structural evidence of metastases on ultrasound and CT scan [73,74].

### 2.5. Management of Persistent or Recurrent Disease Medullary Thyroid Carcinoma

Persistent or recurrent MTC includes a biochemical incomplete response or recurrence and a structurally incomplete response or recurrence [35,52,72]. A biochemical incomplete response is characterized by a high or rising serum level of serum Ctn after at least 3 months post-surgical resection without anatomical evidence of the disease [35,52,68]. Factors that influence treatment selection in patients with persistent or recurrent MTC are the nature of the symptoms, sites of the disease, burden of disease, levels of serum Ctn and CEA, and rate of progression of structural disease [35]. Local or systemic treatment is not advised in patients in whom serum Ctn and/or CEA levels remain elevated following surgery but there is no evidence of structural disease [52,68]. Patients with persistently elevated serum Ctn and/or CEA following total thyroidectomy with prophylactic or therapeutic lymphadenectomy without evidence of structural disease should undergo surveillance with measurement of serum calcitonin and CEA levels every 3–6 months to determine doubling times and an ultrasound of the neck every 6–12 months [35,38,68]. However, additional investigations are indicated if the Ctn or CEA doubling time is less than 24 months, as this is suggestive of an aggressive disease [35,51,52].

Patients with biochemical incomplete responses need to be investigated, and the extent of the investigation depends on the level of serum Ctn [35,51,52]. While neck ultrasound alone is adequate if the level of serum Ctn is not above 150 pg/mL, patients with MTC in whom the level is higher require additional investigations like a CT, MRI, laparoscopy, and bone scan [72]. Patients with structurally residual MTC are categorized as having either stable or unstable disease, and those with stable disease should undergo active surveillance with monitoring of serum levels of Ctn and CEA with imaging by at least an ultrasound every 3–6 months [35,52]. A repeat lymph node dissection followed by external beam radiotherapy (EBRT) or intensity-modulated radiation therapy should be considered in patients with resectable residual in the neck [35]. Furthermore, patients with unstable but resectable disease may also be considered for surgical resection followed by EBRT [35]. In patients with unstable and irresectable disease, as evidenced by worsening symptoms and RECIST-confirmed progression, treatment modalities like radiofrequency ablation, TKIs, cytotoxic chemotherapy, and radio-immunotherapy may be used [35,52,68,72].

## 3. Challenges Associated with the Management of Medullary Thyroid Carcinoma

Medullary thyroid carcinoma is a complex and highly heterogeneous disease. The complexity of MTC spans the entire continuum from its presentation, diagnosis, staging, risk stratification, management, and follow-up of a patient following treatment [21,34,49,56,75,76]. Some of the complexities are generic and applicable to all patients with MTC regardless of the subtype, while others are specific to hereditary or sporadic MTC. Generic challenges linked to the evaluation of suspected or confirmed MTC include the value of routine measurement of the serum Ctn level during the workup of nodular goiter, variable cut-off levels of serum Ctn used for diagnosis and to guide management, the inability of the TNM to accurately quantify the amount of the disease, and the justifiability of thyroid lobectomy as a definitive treatment in patients with low-risk sporadic MTC [71]. The appropriateness of a potentially debilitating surgery for locally advanced or metastatic disease is another challenge. A study by Liu et al. (2024) showed that some patients with metastatic MTC might benefit from resection of the primary tumor, but there is not yet guidelines for selection of individuals for whom it is appropriate [77]. Other problems include how to tailor treatment and follow-up of patients with Ctn-negative MTC. Another dilemma is patients whose serum Ctn level is persistently elevated after curative surgery without structural evidence of residual or recurrent disease.

### 3.1. Serum Calcitonin

Normal serum Ctn level is less than 10 pg/mL, and a level above 100 pg/mL is diagnostic of MTC [78,79]. In a significant proportion of patients with MTC, the serum level is in the “gray-zone” of 10–100 pg/mL, as it may be seen in healthy individuals or patients with benign diseases like chronic thyroiditis, hyperparathyroidism, or kidney dysfunction [79]. The level of serum Ctn is also higher in healthy adult males compared to females and children [65]. The tumor in some patients with MTC may not secrete high levels of Ctn, even if the tumor stains positive on immunohistochemistry. Occasionally, serum levels of both Ctn and CEA may be normal [80,81,82]. In 2015, the ATA task force could not reach a consensus on the utility of serum Ctn level in the general workup of a patient with nodular disease of the thyroid gland [65]. Trimboli et al. investigated the usefulness of routine serum Ctn levels testing during nodular goiter diagnostic workup to exclude MTC over 14.5 years [83]. Predetermined Ctn reference ranges were stipulated for the risk of MTC and included 10–20 pg/mL, 20–100 pg/mL, and greater than 100 pg/mL. The levels of serum Ctn in 170 of the patients were in the upper limit of normal, and 50 underwent thyroidectomy. The final histology in most patients who had elevated serum levels of Ctn showed benign disease, a few of them had PTC, and none had confirmed MTC. Preoperative FNAC is diagnostic in less than 60% of patients with MTC [84,85,86,87]. Trimboli et al. (2022), in a meta-analysis spanning over 10 years, found that FNAC on its own accurately detected MTC in 56.8% of patients and subsequently recommended that FNAC must be utilized in combination with other tools to improve its diagnostic value. The types of cells that constitute MTC are variable and have been described as epithelioid, plasmacytoid, and spindle shape, among others, which may explain the high rate of misdiagnosis [84,88].

### 3.2. Hereditary MTC

Decisions regarding the management of patients with MEN 2B are less complex if the diagnosis is made timeously and the tumor is still intra-thyroidal, as it would be limited to a search for co-existent pheochromocytoma [72,89]. The other decision involves the appropriate time to perform a total thyroidectomy and the extent of a lymphadenectomy [90]. Unlike other manifestations of MEN 2B that are expected in all the affected individuals, phaeochromocytoma develops in half of the cases [89]. The genomic landscape of MEN 2B is uniform and only involves the germline mutation of the *RET* Met918Thr. Of concern is that mutations that drive hereditary MTC in patients with MEN2B occur de novo in some cases, and some patients have no family history and are the first to be affected [91]. Although MTC in patients with classical MEN 2A is less aggressive compared to FMTC- and MEN 2B-associated disease, it is complex to manage as it is heterogeneous with a vast mutational landscape and variable clinical presentations [69,90,91,92,93]. The categorization of hereditary MTC into moderate, high, or highest risk based on the codon of the *RET* proto-oncogene mutated is relevant when the tumor is early, but not when it is advanced, as the outcome is similar for advanced stages of the disease [94]. The occurrence of additional manifestations of MEN 2A, such as hyperparathyroidism and phaeochromocytoma, is variable, which may lead to it being mistaken with sporadic MTC [72,90,95]. The appropriate time to perform a thyroidectomy for the variants of MTC in patients with MEN 2A also varies, and the sequencing of surgery for associated endocrinopathies is also essential [90].

### 3.3. Sporadic MTC

Sporadic MTC is more unpredictable than the MEN 2A- and MEN 2B- associated variants [53,69,84]. For still unknown reasons, some cases of sporadic MTC may remain indolent and associated with a higher 10-year survival despite distant metastases [25]. Serum level of Ctn is not always useful, as it is normal or marginally elevated in some patients with sporadic MTC [96,97]. Most patients with sporadic MTC present like other thyroid tumors and undergo the same diagnostic tests, which include s-TSH, ultrasound, and FNAC. Ultrasound and FNAC are, however, less helpful in the diagnostic workup of sporadic MTC when compared with other malignancies of the thyroid gland except FTC [87]. Thyroid nodules in sporadic MTC may resemble benign disease on ultrasound, which may lead to an erroneous decision to prescribe serial observation or offer an inappropriate initial surgery [98]. The staging of sporadic MTC mirrors that of PTC, although stage-for-stage, MTC is more aggressive and is responsible for a disproportionate number of deaths due to thyroid cancer, despite its relative rarity [69]. A sporadic MTC, even when its maximum diameter is less than 1 cm, is likely to have metastasized to lymph nodes or systemically [53]. The current TNM/AJCC staging system for differentiated thyroid cancer is therefore often not adequate, as it often underestimates the extent and complexity of sporadic MTC [17]. Strategies that have been used to risk-stratify patients with sporadic MTC include segregating the tumors using the levels of Ctn and/or other markers, such as CEA, procalcitonin, pro-gastrin-releasing peptide, and carbohydrate antigen 19.9 (CA 19.9) [53,71]. Other measures include the detailed analysis of ultrasound images and other imaging findings, such as CT results, FNAC slides, and genetic, epigenetic, proteomic, and metabolomic landscapes [99,100,101].

### 3.4. Targeted Therapy

Management options for stable locally advanced or metastatic disease include surgery, external beam radiotherapy, radiofrequency, ablation, and stereotactic radio-guided surgery [43,46]. The surgical resection of locally advanced or metastatic MTC is, however, inappropriate if it leads to significant postoperative complications, such as bilateral recurrent laryngeal nerve injury and permanent hypoparathyroidism without survival benefit [55,102]. The other options for advanced MTC are targeted therapy with tyrosine kinase inhibitors [44,48,100]. Targeted immunotherapies are expensive and may lead to significant side effects, such as nausea, diarrhea, hypertension, bleeding, thrombosis, skin changes, and weight loss [44].

## 4. Artificial Intelligence in the Healthcare Industry

The use of machines to simulate human actions was first proposed by Alan Turing in 1950, and John McCarthy introduced the term artificial intelligence in 1956 [103]. The subfields of AI include machine learning (ML), artificial neural networks (ANN), and deep learning (DL) [104]. In ML, machines are trained to accomplish what historically could only be performed by a human being, whereas ANN and DL are more complex and require advanced computing. Various algorithms are used for the classification of features in ML, including logistic regression (lR), random forest (RF), decision tree (DT), support vector machine (SVM), k-nearest neighbors (KNN), gradient-boosting machine (GBM), and extreme gradient boosting (XGBoost) [101]. Data extraction and classification may be fully supervised, partially supervised, or unsupervised. Using computer software, variables that are deemed relevant are selected by experts and fed into the ML algorithms to develop a prediction model. The comparison of the performance of different AI models is based on their accuracies and values of the area under the receiver operating characteristic curve. In DL, computers extract, analyze, and interpret quantitative features without supervision [104]. Deep learning is more complex than ANN, as it has multiple hidden layers [105,106,107,108,109]. For the arrangement of AI options according to levels of complexity, see Figure 1.

The adoption of AI in the healthcare industry has been slow, and its use in diagnostic investigation and the risk stratification, management, and follow-up of patients is limited [110,111]. The use of AI to assist in decision-making is useful in managing cancer, as cancer is a heterogeneous disease, the heterogeneity and complexity of which are often missed by the traditional methods of staging the disease [110,112,113]. The interpretation of imaging and cytological or histological findings in a patient suspected to have MTC relies on expertise, which is not universally available [104,111,114]. Data collected during the evaluation of patients with suspected MTC include family history and lifestyle, exposure to environmental risk factors, the results of imaging investigation(s), and the histological findings, genomics, epigenomics, proteomics, and metabolomics of a cancer [115,116,117,118,119,120,121]. Incorporating AI allows for the extraction of a vast amount of quantitative information and integrating them for decision-making.

Several authors have reported on the usefulness of radiomics [122], pathomics [123], genomics [69,124], and other omics in the management of thyroid cancers [121]. ML and the convolutional neural network (CNN) version of DL are utilized extensively in other branches of medicine that rely on imaging and the evaluation of pathological specimens for screening, diagnosis, staging, risk stratification, treatment selection, and follow-up [101,109,118]. The commonly used ML algorithms for computer-aided decision-making the healthcare industry include SVM, RF, GBM, DT, KNN, Bayesian networks (BN), lean six sigma (LSS), and natural language processing [NLP] [101,125,126,127,128,129,130,131,132,133,134,135,136]. The benefits of using AI during the evaluation and management of patients with MTC include the enhancement of the ability to distinguish MTC from other causes of goiter, standardizing procedures for sample collection, and data integration to reduce variability [136]. Some AI algorithms can classify MTC patients based on genetic mutations and protein expression profiles or predict patient outcomes and treatment responses based on clinical, genetic, and proteomic data [128,129]. Additionally, AI can integrate and simultaneously analyze data from laboratory and imaging investigations performed during the evaluation and management of patients with MTC [130,133,135]. Table 1 contains lists of various models of AI algorithms available for use during computer-aided decision-making in the management of patients with MTC, including their application, function, and potential benefits.

Several studies have proven the beneficial role of AI in the diagnostic workup and management of various diseases, including thyroid cancer [107,108,111,114,115,118]. Most studies on the role of AI in thyroid cancer focused solely on PTC. The rarity of MTC and the limited availability of expertise may not allow for the accumulation of enough data for the training and testing of AI models [118,124]. However, fundamental to the appropriate management of MTC is a diagnosis when the tumor is small, the exclusion of lymph nodes and distant metastases, determining the completeness of surgical resection, and the timely detection of recurrence. The other key issues are establishing the genomic variant and molecular subtype of MTC, especially if the tumor is locally advanced or metastatic and treatment with target therapy is being considered.

## 5. Application of AI during the Investigation of Medullary Thyroid Carcinoma

The majority of cases of MTC present as thyroid nodules, and thyroid nodules are found in around 7% of the general population following clinical examination and up to 70% following ultrasound examination [136]. Most thyroid nodules are benign and do not require surgery unless they are toxic or cause compression symptoms [137]. Using a computer program to extract and subsequently combine demographic data, lifestyle, environmental factors, clinical findings, ultrasound results, and FNAC can assist in differentiating a benign thyroid nodule from a malignant thyroid nodule [104,105,108,117].

### 5.1. Radiomics

Radiomics is the process of extracting quantitative information from imaging investigations beyond the capability of a human eye. Any imaging investigation, including an ultrasound, CT scan, MRI, or radioisotope scan, is amenable to the use of AI to extract quantitative data. The results of radiomics alone or combined with clinical information and other omics (holomics) are useful for diagnosis, risk stratification, and prognostication in patients with cancer and other diseases [109,126]. Radiomics may be applied following any imaging investigations, including ultrasound, CT scan, MRI, and PET/CT [100,109]. The six main steps in radiomics are the acquisition of an image, processing, marking areas of interest (segmentation), the extraction of key features, the development of a predictive model, and verification [109,138]. The selection of area of interest may be performed by experts in the field or automatically by the computer. Similar steps are followed during the development of training, validation, and testing models. Imaging results for training and validation may be sourced from archived results.

The interpretation of findings following an ultrasound examination of the thyroid gland is influenced by the experience of the sonographer [104,116,137]. Features that are suggestive of malignancy and are assessed during an ultrasound evaluation of a thyroid nodule include whether a nodule is cystic or solid and the echogenicity, vascularity, existence of calcifications, even-ness of its border, shape, and stiffness [78,87,139]. An MTC may however appear benign or resemble PTC on ultrasound [32]. Additionally, changes critical for the diagnosis of MTC may not be visible to the naked eye. Adding an AI-aided analysis of thyroid nodules enables a detailed evaluation of the nodule and outperforms less experienced radiologists in differentiating benign from malignant lesions [104,111,117,137]. An artificial intelligence-guided analysis of ultrasound images can differentiate MTC from PTC. Whereas PTC that is less than 1 cm (microcarcinoma) maximum in low-risk patients may be observed, the same cannot be said of MTC, as it may be hereditary or harboring aggressive features.

The integration of demographic information, clinical findings, and ultrasound and CT scan results using AI technology can more accurately differentiate benign from malignant nodules in patients whose FNAC result is indeterminate [105,106,140]. Another potential benefit of digital imaging, whole-slide imaging, and the application of AI is the ability to non-invasively identify the underlying mutation driving the cancer [141]. Radiomics may also be repeated after the administration of contrast or during follow-up after the initiation of treatment; the so-called delta radiomics can improve the diagnostic ability of an investigation or guide the timeous adjustment of therapy [142]

Although not yet explicitly utilized in MTC, an AI-guided analysis of findings from other imaging investigations can be used to characterize a thyroid nodule following Bethesda III or IV FNAC results and predict the probability of metastasis to the lymph nodes and predict the mutational status of the tumor [105,140,143]. For example, AI-guided interpretation of CT scan findings is more accurate than radiologists in distinguishing benign from malignant thyroid nodules with indeterminate cytology results [118]. Using AI technology allows for the segmentation of lesions and the mining of a vast amount of quantitative information beyond what a human can accomplish, regardless of experience, even when working in a multidisciplinary team [140]. A dual-energy CT scan is among the simple versions of AI-guided analysis that is useful for differentiating benign from malignant lesions in thyroid nodules in which the FNAC result is indeterminate [144].

### 5.2. Pathomics

The interpretation of FNAC specimens of suspected MTC may be complex and require experience. It is not uncommon for MTC to be mistaken for PTC, FTC, poorly differentiated thyroid carcinoma (PDTC), anaplastic thyroid carcinoma (ATC), lymphoma, and benign conditions [114]. The ability to create digital slides and whole-slide imaging allows for the application of the AI-aided diagnosis of thyroid cancer [123,145]. The sequence followed during pathomics includes the digital conversion of cytology or histopathology slides, whole-slide imaging, supervised or unsupervised, of the region of interest, analysis, and classification [138,145]. The steps that are followed in radiomics are like those used during pathomics. Similarly, the first step is the acquisition of an image using an ultrasound, CT scan, MRI, or PET/CT, followed by the segmentation and selection of the region of interest [141]. Ideally, samples for both pathomics and radiomics should be divided into training, testing, and validation sets. Radiomics may be repeated during follow-up with a patient after treatment. For a comparison of the steps that are followed during radiomics and pathomics, see Figure 2.

The ability to share the slides for a second opinion and segmentation of the slides with the selection of the region of interest is likely to lead to a more detailed assessment of a tumor and its environment, leading to accurate preoperative grading of cancer [101,104,146,147,148]. The artificial intelligence-aided analysis and interpretation of FNAC can accurately diagnose the subtype, differentiation, grade, and genomic profile of a cancer and predict the likelihood of metastasis and recurrence [113,145].

### 5.3. Epigenomics

The application of epigenomics in a workup of thyroid cancer is firmly established. Epigenetics include histone modification and alteration of the expression of lnRNA and miRNA. Under- or over-expression of miRNAs is associated with the development and progression of MTC in some patients [30,106,149,150,151,152,153]. The levels of some miRNAs, including miR-34a, miR-144, and miR-375, can distinguish MTC from other causes of goiter [30,149,150,151,152]. Some of the miRNAs that are potentially useful for distinguishing hereditary from sporadic MTC include miR-9, miR-183, and miR-375 [30,149,150,153]. Additionally, a change in the level of expression of some miRNAs, including miR-21, miR-127, miR-224, miR-375, and miR-597, may be useful for the risk stratification of patients with MTC, including the prediction of lymph node metastasis. The inclusion of the pattern of change in miRNA expression may potentially improve decision-making during the treatment of patients with MTC [5,30,148,149,150,151]. For a summary of changes in the levels of miRNA seen in patients with MTC, see Table 2.

### 5.4. Other Omics for the Investigation and Management of Cancer

Other AI-assisted diagnostic strategies shown to be useful in patients with other malignancies of the thyroid, including PTC, are proteomics, metabolomics, glycomics, and lipidomics [31,32,105,112,154,155,156]. Among the processes involved in the post-translational modification of proteins is glycosylation, which involves the addition of carbohydrate chains [145]. Proteins that are glycosylated may be intracellular or extracellular. A change in the levels of glycosylation is seen in several cancers, including thyroid cancers. Calcitonin, CEA, and CA 19.9 are among the glycosylated proteins that are relevant for the screening, diagnosis, and treatment of MTC. A change in the expression of genes that code glycosylated proteins, epigenetic modification, or the glycans themselves may lead to the development of cancer and influence its progression or response to treatment [157]. A study of the pattern of change in the glycosylated products (glycomics) in the tumor, TME, or blood is useful for screening, diagnosis, the selection of treatment, and the prediction of response to treatment and the likelihood of recurrence in patients with various cancers, including thyroid cancer.

The analysis of fluid (fluidomics), such as blood, saliva, or ascitic fluid, for molecules other than Ctn and CEA may provide valuable information for the screening, diagnosis, risk stratification, and follow-up of patients with MTC. Among the predisposing factors of cancer is chronic inflammation, which does not cease to be active even after the cancer has developed but continues and has an influence on the progression, metastatic potential, and recurrence of the tumors [158]. Markers of inflammation that are commonly monitored include the neutrophil-to-lymphocyte ratio (NLR), lymphocyte-to-monocyte ratio (LMR), mean platelet volume (MPV), and platelet distribution width (PDW). There are conflicting reports on the usefulness of NLR, LMR, and PDW for the prediction of the local spread, lymph node metastasis, and the possibility of biochemical cure of MTC [159,160].

Combining the level of inflammatory markers with results from genomics, epigenomics, metabolomics, and other omics may further enhance decision-making during the management of MTC. There is still a paucity of evidence regarding the use of the other omics other than proteomics, metabolomics, radiomics, or pathomics during the investigation of patients with confirmed or suspected MTC [28,29,138,161]. The results of a study by Jajin et al. using gas chromatography-mass spectrometry-based untargeted metabolomics showed that MTC is associated with a change in the plasma levels of amino acids and lipid metabolites, among others [154]. They found that patients with MTC had lower levels of leucine than healthy individuals. The use of AI can improve diagnostic accuracy for MTC by analyzing imaging data, integrating multiple data sources, including imaging, genetic, and clinical, and identifying subtle features indicative of MTC. The other benefits of AI-aided decision-making are accurate staging and better characterization of the disease, including its genomic landscape, risk stratification, prediction of treatment response, and timeous detection of progressive disease [101,162,163,164,165]. Data from the different omics can be integrated and analyzed simultaneously for individualized management of a patient with MTC [165]. Using AI may improve access to expertise by patients and healthcare workers in low- and middle-income countries by using, for example, mobile devices and whole-slide imaging and digital slides [166]. Table 3 is a summary of the potential applications of AI in patients with MTC using a combination of findings from laboratory, imaging, and genetic analysis.

## 6. Application of AI in MTC

Most patients with sporadic MTC have metastases to central and or lateral cervical lymph nodes at presentation and are not cured if treated only with a lobectomy or thyroidectomy [100,168]. The size of the primary tumor and serum level of Ctn are not always useful for predicting central, ipsilateral, and/or contralateral lateral cervical lymph node metastasis [61,169]. The application of AI facilitates the structuring of big data and the analysis and integration of findings from various investigations for improved quantification of the burden of the disease, risk stratification, and tailoring of treatment of MTC (101). Guo et al., 2023, in a study involving the records of 2049 patients from the publicly available Survival, Epidemiology, and End Results Reporting (SEER) Database, found that ML algorithms could accurately predict the existence of distant metastases in patients with MTC [101]. The study showed that the RF model outperformed other ML algorithms in the prediction of metastases, with an AUC above 0.8. The model found that a combination of male gender, age above 55 years, size greater than 4 cm, multifocality, extra-thyroidal extension, and lymph node metastases increased the likelihood of distant metastases in patients with MTC.

### 6.1. Metastatic Workup of Medullary Thyroid Carcinoma

A study by Zhang et al. (2024) involving patients with early MTC found that AI algorithms combining clinical and ultrasound findings accurately predicted the existence of metastatic cervical lymph nodes [168]. Similarly, Li et al., (2020) reported that the AI-enabled extraction of quantitative data from ultrasound images of thyroid cancer can accurately predict the involvement of regional lymph nodes [105]. Zhang et al. (2023) used ML algorithms to develop a nomogram for early detection of distant metastases in patients with MTC to address the unreliability of symptoms and serum Ctn and CEA levels for monitoring following surgery [81]. The abovementioned study utilized 1901 records of patients with MTC from the SEER database for training and testing, while 111 records for validation were from their hospital, the First Hospital of Jilin University. The study used RF, LR, GBDT, and SVM to select features that were predictive of distant metastasis. Key findings from Zhang et al. (2023) included that RF performed the best, with an AUC of 0.8786 (95%CI, 0.870–0.9503), and that a combination of the patient’s age, larger tumor size, non-total thyroidectomy, and involvement of the cervical lymph node was predictive of distant metastasis [81]. Computed tomography-based AI models can also accurately predict metastasis to the central and lateral cervical lymph nodes in patients with PTC [105]. Large tumor sizes greater than 4 cm [169], high levels of Ctn, high CEA [18,100,101], high CA19.9 [53,115], high-grade tumors [115], tumors with areas of desmoplastic reaction [44,170] or lympho-vascular invasion [169], and/or RET proto-oncogene mutations increase the likelihood of distant metastasis [171]. However, distant metastasis may be small and missed during preoperative investigations, only to be suspected when the level of Ctn does not normalize or become not detectable following curative surgery [72]. The use of various ML and CNN programs combining personal tumor location and size and ultrasound findings can accurately predict the existence of distant metastasis [106,107,108,163].

### 6.2. Risk Stratification of MTC

The TNM/AJCC, MTC grading system and serum levels of Ctn and CEA do not sufficiently risk-stratify MTC [4,15,17]. Furthermore, MTC is often missed during preoperative investigations and diagnosed following diagnostic lobectomy or inappropriate surgical procedures, which may negatively affect the outcome [35,71]. Locally advanced or metastatic MTC may be, in some patients, unexpectedly indolent and not warrant aggressive and sometimes debilitating surgery [25]. Guo et al. (2023), in a retrospective study based on the records of patients who had MTC, demonstrated that AI can accurately predict the existence of cervical lymph node metastasis in patients with clinically and radiological node-negative disease.

### 6.3. Treatment of Locally Advanced and Metastatic MTC

The application of AI can guide the extent of thyroidectomy and lymphadenectomy in patients with MTC [168]. The available treatment options are not uniformly suitable for all patients with MTC and are dependent on the anatomical extent of the disease, serum level of Ctn and CEA, and genomic landscape of the tumor, among others [172,173,174,175,176]. Genetic and molecular analyses are expensive and are not universally available. The mutational landscape and molecular profile can sometimes be determined following a liquid biopsy, a virtual biopsy using ultrasound, other imaging modalities [99], and a cytopathological analysis [113,139,164]. Findings of genomic studies with or in combination with radiomics, proteomics, and metabolomics may support the diagnosis of MTC [104,160,165,177]. Table 4 contains a summary of possible combinations of omics that could be combined and simultaneously analyzed to improve accuracy in the diagnosis, staging, risk stratification, and follow-up of patients with MTC.

There is a need for the development of a risk stratification system that would simultaneously integrate and harmonize findings from demographic and clinical assessment [22,108,115], imaging [121], histopathology or cytopathology [29,113,126], genomics [29,100], epigenomics [31], proteomics and metabolomics [32,101,117], and fluidomics [4] to guide decision-making to enable the appropriate management of patients with MTC.

Among the limitations of the AI-aided management of MTC is restricted availability due to high-cost infrastructure for modern computing, especially in low- and middle-income countries. Developing algorithms for training and testing AI models requires expertise and an adequate sample size for training, validation, and testing, which may be a challenge as MTC is rare, and there is not enough expertise, especially in LMIC [179]. Small sample sizes for use in developing a training or validation process may lead to over-fitting and lead to a generation of models that do not accurately reflect the actual situation. The rarity of MTC, however, encourages worldwide collaboration that would augment the already available data. Another strategy that can be used to mitigate the small sample size and limited availability of validation samples includes the so-called transfer learning, during which findings from complementary tests are combined and fed into ML algorithms [180]. The inability to scientifically explain some of the decisions after applying ML algorithms, the so-called “blackbox”, is common to all AI programs. Among the concerns of patients regarding the use of AI is the potential violation of privacy by making personal information accessible to the public [181].

## 7. Conclusions and Perspectives

Medullary thyroid carcinoma (MTC), despite its rarity and stable prevalence, poses a significant challenge due to its aggressive nature and high mortality rate compared to other thyroid malignancies. Its heterogeneity, whether hereditary or sporadic, complicates treatment approaches not adequately addressed by guidelines tailored for papillary thyroid carcinoma (PTC). The extensive data collected during MTC investigation and management spanning clinical, radiological, pathological, mutational, and immunological domains overwhelms manual analysis and integration capabilities. To overcome this, integrating artificial intelligence (AI) and holomics emerges as a pivotal solution. AI, in conjunction with holomics, offers a transformative approach across MTC’s diagnostic, staging, risk stratification, management, and follow-up phases. By swiftly processing diverse data types and uncovering nuanced patterns, AI enhances precision in treatment planning beyond human capability. Furthermore, this approach fosters global collaboration by synthesizing worldwide clinical experiences, thereby enriching understanding and refining therapeutic strategies for MTC.

The future of AI in MTC investigation is promising, with continued advancements in AI algorithms and the growing availability of comprehensive holomics datasets. Collaborative efforts among researchers, clinicians, and AI experts will develop and validate tailored AI tools. The development of interpretable AI models will be crucial for clinical acceptance. Expanding AI applications to other areas of thyroid cancer research, such as risk stratification and the discovery of novel therapeutic targets, holds great potential for improving patient outcomes. AI will play an increasingly integral role in the investigation and management of MTC, thereby transforming the landscape of thyroid cancer care.

Moreover, deploying AI aligns with Goal Number 3 of the United Nations Sustainable Development Goals (SDG) for 2030, aiming to ensure universal access to personalized healthcare. This synergy underscores AI’s potential to revolutionize MTC management, enhancing survival rates and quality of life globally. In summary, integrating holomics and AI into MTC management represents a paradigm shift in precision oncology. This innovative approach addresses the unique challenges of MTC, promotes global healthcare equity, and promises improved outcomes for patients worldwide. As AI continues to evolve, its synergy with holomics holds promise for personalized, effective, and accessible care, setting a new standard in oncological practice.

## Figures and Tables

**Figure 1 cancers-16-03469-f001:**
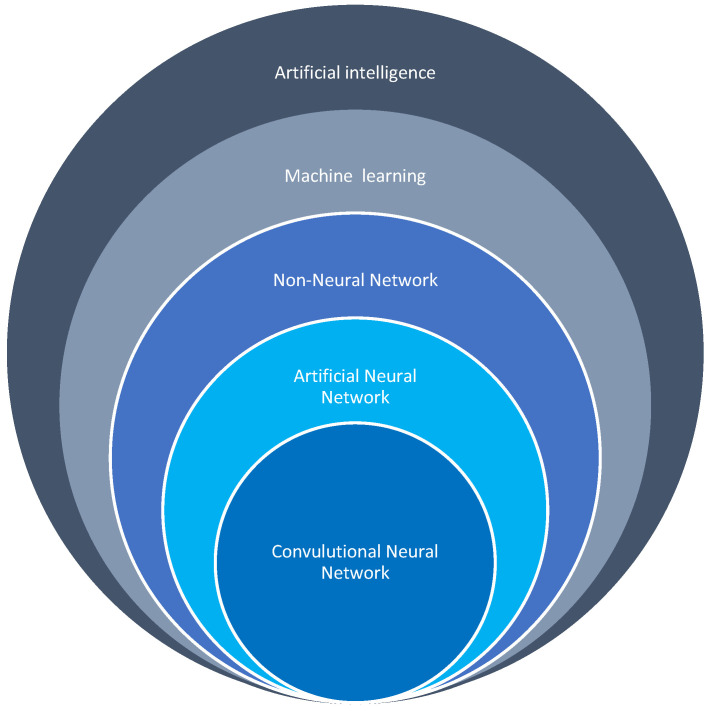
Categories of AI processes arranged according to their complexities.

**Figure 2 cancers-16-03469-f002:**
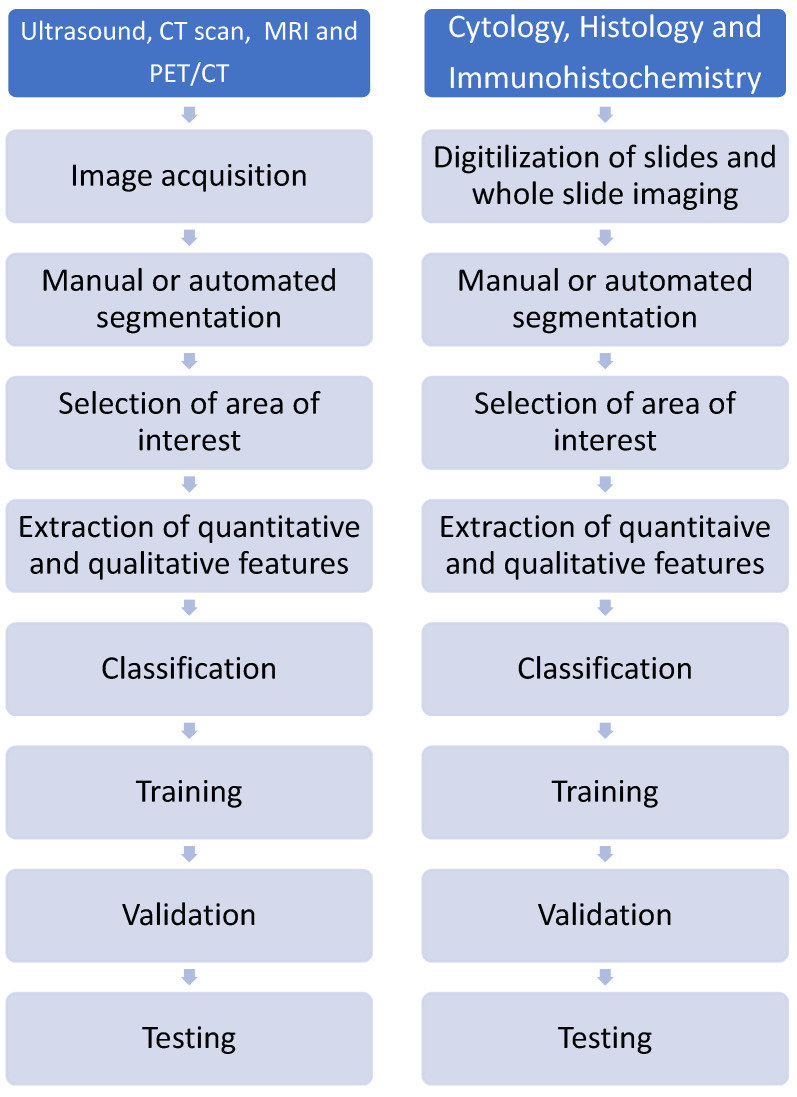
Comparison of steps followed during development of training and validation models in radiomics and pathomics.

**Table 1 cancers-16-03469-t001:** List of AI intelligence options for processing and integrative analysis of tumors.

Algorithm	Application	Function	Example
CNN [125,126]	Image analysis (ultrasound, CT, MRI)	Automatically identify features indicative of MTC	Distinguish between benign and malignant thyroid nodules with high accuracy
SVM [127]	Classification of thyroid lesions	Find the optimal hyperplane that separates different classes of data points	Classify MTC patients based on genetic mutations and protein expression profiles
RF [128]	Classification and regression	Build multiple decision trees and merge them to improve predictive accuracy	Predict patient outcomes and treatment responses based on clinical, genetic, and proteomic data
KNN [129]	Classification and regression	Classify a data point based on the classification of its neighbors	Classify thyroid nodules based on ultrasound features to determine the likelihood of malignancy
ANN [130]	Complex pattern recognition	Process data in layers to learn intricate patterns	Integrate multi-omics data to predict disease progression and patient survival
GBM [131]	Classification and regression	Build models sequentially, each correcting errors of the previous ones	Predict recurrence risk in MTC patients by analyzing clinical and molecular data
RNN [130]	Time-series prediction and sequential data analysis	Maintain a memory of previous inputs to predict future outcomes	Analyze longitudinal patient data to predict future disease progression and treatment outcomes
Autoencoders [132]	Data dimensionality reduction and feature extraction	Compress data into a lower-dimensional representation and reconstruct it back	Identify key features in genetic and proteomic data that are most indicative of MTC
BN [130]	Probabilistic inference and decision-making	Represent variables and their conditional dependencies through directed acyclic graphs	Model relationships between genetic mutations, environmental factors, and MTC development
NLP [133]	Process and analyze unstructured clinical texts	Extract relevant information from EHRs, pathology reports, and the scientific literature	Extract patient data and clinical outcomes related to MTC, integrating with omics data for comprehensive analysis
Geolocation [134]	Epidemiology and public health planning	Mapping the geographical distribution of MTC cases to identify environmental and genetic risk factors; planning targeted screening programs and resource allocation	Identifying regions with higher incidence rates of MTC to implement targeted screening programs and allocate resources effectively; correlating regional dietary habits and environmental exposures with MTC incidence
Survival Analysis [135]	Prognostic predictions and patient stratification	Estimating time until events (disease progression, recurrence, death) and identifying prognostic factors	Developing risk stratification models based on clinical, genetic, and demographic variables to predict patient outcomes and tailor follow-up and monitoring strategies
Lean Six Sigma [136]	Process optimization and efficiency in clinical workflows	Streamlining clinical processes, reducing diagnostic errors, and improving treatment workflows by eliminating inefficiencies	Standardizing procedures for sample collection and data integration to reduce variability and improve the reliability of holomic analyses; ensuring consistent follow-through on diagnostic and treatment protocols

**Table 2 cancers-16-03469-t002:** Change in miRNA expression in medullary carcinoma and the implications.

Name of miRNA	Expression	Consequences	References
miR-375	Overexpressed	Diagnosis, lateral lymph nodes predicted, worse prognosis. Distinguishing hereditary from sporadic MTC.	[30,149,150]
miR-127	Underexpressed	Aggressive sporadic disease.	[149,151]
miR-429	Overexpressed	Not yet specified	[149]
miR-592	Overexpressed	Poor prognosis.	[106]
miR-224	Underexpressed	Poor prognosis	[30]
miR-199-5p	Underexpressed	Not yet specified	[149]
miR-199a-3p	Underexpressed	Not yet specified	[149]
miR-34a	Underexpressed	Biomarker of MTC	[152]
miR-9	Underexpressed	Distinguishing hereditary versus sporadic	[153]
miR-21	Overexpressed	Prediction of lymph node and distant metastasis	[30]
miR-144	Overexpressed	Biomarker of MTC	[152]
miR-183	Overexpressed	Prediction of lateral lymph node involvement, distant metastasis, and high mortality and distinguishing hereditary from sporadic MTC.	[30,153]

**Table 3 cancers-16-03469-t003:** Summary of potential applications and benefits of AI in patients with MTC.

Application	Description	Examples/Impact
Enhanced diagnostic accuracy [101,162]	AI improves diagnostic accuracy for MTC by analyzing imaging data, integrating multiple data sources, including imaging, genetic, and clinical, and identifying subtle features indicative of MTC. Traditional methods like ultrasound and fine-needle aspiration can be inconclusive.	AI-powered image recognition systems distinguish between benign and malignant thyroid nodules more accurately than human radiologists, leading to an early and accurate diagnosis, which is essential for the effective treatment of MTC.
Personalized treatment plans [163]	AI personalizes treatment plans by analyzing genetic and molecular data to identify specific mutations and biomarkers associated with MTC. It predicts patient responses to targeted therapies, optimizing treatment efficacy and minimizing side effects. AI updates treatment recommendations as new data become available.	AI guides the selection of targeted therapies, such as tyrosine kinase inhibitors, ensuring that patients with MTC receive the most current and effective treatments based on their unique genetic profile.
Prognostic predictions [164]	AI develops predictive models to estimate disease progression and patient outcomes by integrating diverse data points like the stage of cancer, genetic mutations, and the patient’s characteristics. Machine learning algorithms analyze historical patient data to identify patterns and risk factors associated with recurrence or metastasis.	AI helps clinicians stratify patients into different risk categories and tailor follow-up and monitoring strategies, providing more accurate prognostic information and improving the long-term management of MTC.
Holomic integration [165]	Holomics integrates various omics data to provide a holistic view of MTC at the molecular level. AI analyzes and interprets complex datasets to identify gene expression patterns, detect protein biomarkers, and analyze metabolic profiles, offering a more complete understanding of the disease.	AI-enabled holomics uncovers novel insights into MTC pathogenesis and identifies new therapeutic targets, leading to better diagnostic and therapeutic strategies.
Comparative insights in HICs versus LMICs [166]	AI application varies between high-income countries (HICs) and low- and middle-income countries (LMICs). HICs benefit from advanced healthcare infrastructure and cutting-edge technologies, while LMICs face challenges like limited resources and insufficient training. AI can bridge these gaps by deploying diagnostic tools via mobile health platforms and optimizing resource use.	AI-driven diagnostic tools enable remote diagnosis and expert consultations in resource-limited settings, making high-quality cancer care more accessible and efficient. This reduces disparities between HICs and LMICs in MTC management.
Future prospects [167]	The future of AI in MTC investigation is promising, with continued advancements in AI algorithms and the growing availability of comprehensive holomic datasets. Collaborative efforts among researchers, clinicians, and AI experts will develop and validate tailored AI tools. The development of interpretable AI models will be crucial for clinical acceptance.	Expanding AI applications to other areas of thyroid cancer research, such as risk stratification and the discovery of novel therapeutic targets, holds great potential for improving patient outcomes. AI will play an increasingly integral role in the investigation and management of MTC, transforming the landscape of thyroid cancer care.

**Table 4 cancers-16-03469-t004:** List of omics options available for use during screening, investigation, management, and follow-up of patients with MTC.

Target	Omics Option	References
Diagnosis	Fluidomics	[4]
	Genomics	[165]
	Glycomics	[162]
	Metabolomics	[160]
	Pathomics	[12,145]
	Proteomics	[177]
	Radiomics	[104,111]
	Transcriptomics	[5]
Staging	Fluidomics	[4,178]
	Metabolomics	[155]
	Radiomics	[111,156,173]
	Pathomics	[146,147]
	Transcriptomics	[5]
Risk stratification	Transcriptomics	[5,12,55]
	Fluidomics	[4,99,173,176]
	Genomics	[94]
	Glycomics	[177]
	Immunomics	[47,174]
	Pathomics	[146,147,156]
	Radiomics	[173]
Selection of treatment	Fluidomics	[176]
	Genomics	[99]
	Immunomics	[47]
	Pathomics	[147]
	Radiomics	[141]
	Transcriptomics	[5,8,12]
Follow-up	Delta radiomics	[143]
	Fluidomics	[4]
	Genomics	[176]
	Metabolomics	[155]

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
