# Peer review of "Holomics and Artificial Intelligence-Driven Precision Oncology for Medullary Thyroid Carcinoma: Addressing Challenges of a Rare and Aggressive Disease"

_cancers, 2024, doi:10.3390/cancers16203469_

Round 1
Reviewer 1 Report
Comments and Suggestions for Authors
This manuscript, titled “Holomics and AI-Driven Precision Oncology for Medullary Thyroid Carcinoma: Addressing Challenges of a Rare and Aggressive Disease”, provides an in-depth and comprehensive review of the current state of medullary thyroid carcinoma (MTC) management, with a particular focus on the integration of holomics and artificial intelligence (AI) to improve patient outcomes. This is a well-written and informative review that provides valuable insights into the potential of holomics and AI in improving the management of MTC. Here are some detailed comments and suggestions:
1. The review covers various aspects of MTC, including its epidemiology, pathogenesis, diagnosis, management, and the challenges associated with its treatment. The structure of the manuscript is logical, with a clear flow from the introduction to the conclusion.
2. While the article mentions various AI algorithms and their potential applications in MTC management, a more focused discussion on specific AI applications and their limitations would be beneficial.
3. The conclusion effectively summarizes the main points and highlights the potential of AI and holomics in MTC management. It would be beneficial to include a section on future research directions and how AI might be further integrated into clinical practice.
4. The authors mention the potential of AI to bridge the gap between high-income and low- and middle-income countries. A more detailed discussion on the challenges and solutions for implementing AI in these low- and middle-income countries would strengthen this point.
5. The use of figures and tables to summarize complex information is effective. However, the font size in Figure 1 is inconsistent, and the resolution of the image and text in Figure 3 are low, the quality of figures could be improved for better clarity. Additionally, the sequence of the figures should be reviewed for coherence. It should be noted that Figure 3 is repeated, which may require correction.
- Clarity and Focus: Ensure that the manuscript maintains a clear focus on the role of AI and holomics in MTC management, avoiding digressions into unrelated areas.
- Data Integration: Provide more detailed examples of how AI integrates various data types to improve MTC management.
- Clinical Examples: Include case studies or clinical trial data that demonstrate the effectiveness of AI-driven approaches in MTC treatment.
- Ethical Considerations: Discuss any ethical considerations related to the use of AI in healthcare, particularly in the context of data privacy and informed consent.
- Future Directions: Elaborate on potential future developments in AI and holomics that could further improve MTC management.
Comments on the Quality of English Language
English editing is recommended.
Author Response
Reviewer 1
- The review covers various aspects of MTC, including its epidemiology, pathogenesis, diagnosis, management, and the challenges associated with its treatment. The structure of the manuscript is logical, with a clear flow from the introduction to the conclusion.
- While the article mentions various AI algorithms and their potential applications in MTC management, a more focused discussion on specific AI applications and their limitations would be beneficial.
- The conclusion effectively summarizes the main points and highlights the potential of AI and holomics in MTC management. It would be beneficial to include a section on future research directions and how AI might be further integrated into clinical practice.
- The authors mention the potential of AI to bridge the gap between high-income and low- and middle-income countries. A more detailed discussion on the challenges and solutions for implementing AI in these low- and middle-income countries would strengthen this point.
- The use of figures and tables to summarize complex information is effective. However, the font size in Figure 1 is inconsistent, and the resolution of the image and text in Figure 3 are low, the quality of figures could be improved for better clarity. Additionally, the sequence of the figures should be reviewed for coherence. It should be noted that Figure 3 is repeated, which may require correction.
Recommendations for Improvement:
- Clarity and Focus:Ensure that the manuscript maintains a clear focus on the role of AI and holomics in MTC management, avoiding digressions into unrelated areas.
- Data Integration:Provide more detailed examples of how AI integrates various data types to improve MTC management.
- Clinical Examples:Include case studies or clinical trial data that demonstrate the effectiveness of AI-driven approaches in MTC treatment.
- Ethical Considerations:Discuss any ethical considerations related to the use of AI in healthcare, particularly in the context of data privacy and informed consent.
- Future Directions:Elaborate on potential future developments in AI and holomics that could further improve MTC management.
|
Reviewer 1 |
|
|
Recommendation Response |
Response |
|
· 1. The review covers various aspects of MTC, including its epidemiology, pathogenesis, diagnosis, management, and the challenges associated with its treatment. The structure of the manuscript is logical, with a clear flow from the introduction to the conclusion. |
1. Noted. Thank you. We sincerely appreciate the message. |
|
2. While the article mentions various AI algorithms and their potential applications in MTC management, a more focused discussion on specific AI applications and their limitations would be beneficial. · |
2. Noted. We tried to briefly touch of the commonly used machine learning (AI) algorithms in Table 1 and Table 3. We however tried to avoid losing focus on the main purpose of the manuscript, which was to introduce potential use of AI to enhance decision making during management of patients with MTC. |
|
3. The conclusion effectively summarizes the main points and highlights the potential of AI and holomics in MTC management. It would be beneficial to include a section on future research directions and how AI might be further integrated into clinical practice. 4. The authors mention the potential of AI to bridge the gap between high-income and low- and middle-income countries. A more detailed discussion on the challenges and solutions for implementing AI in these low- and middle-income countries would strengthen this point. · |
3. Noted.
|
|
5. The use of figures and tables to summarize complex information is effective. However, the font size in Figure 1 is inconsistent, and the resolution of the image and text in Figure 3 are low, the quality of figures could be improved for better clarity. Additionally, the sequence of the figures should be reviewed for coherence. It should be noted that Figure 3 is repeated, which may require correction. · |
5. Noted. Thank you. We have deleted both Figure 1 and Figure 3. |
|
· 6. Clarity and Focus: Ensure that the manuscript maintains a clear focus on the role of AI and holomics in MTC management, avoiding digressions into unrelated areas. · |
6. Noted. Thank you. We have reviewed the manuscript and made changes in some areas (see markings in yellow). |
|
· 7. Data Integration: Provide more detailed examples of how AI integrates various data types to improve MTC management. · |
7. Thank you for the comment. We have tried to cite some examples and specific machine algorithms that would be useful for regression analysis and classification in patients with MTC in Table 1 and Table 3. |
|
· 8. Clinical Examples: Include case studies or clinical trial data that demonstrate the effectiveness of AI-driven approaches in MTC treatment.
|
8. Thank you very much for the comment. Although we fully agree there is as yet no clinical trial on the role of AI-guided decision making in MTC. More work has been done on papillary thyroid cancer and other malignancies beyond the thyroid gland. Adoption of AI is relatively new especially by the medical profession. |
|
· 9. Ethical Considerations: Discuss any ethical considerations related to the use of AI in healthcare, particularly in the context of data privacy and informed consent.
|
9. Thank you. We have added a statement on ethical considerations. |
|
· 10. Future Directions: Elaborate on potential future developments in AI and holomics that could further improve MTC management.
|
10. Thank you. We have added a statement on future perspectives under the conclusion. “The future of AI in MTC investigation is promising with continued advancements in AI algorithms and the growing availability of comprehensive holomic datasets. Collaborative efforts between researchers, clinicians, and AI experts will develop and validate tailored AI tools. The development of interpretable AI models will be crucial for clinical acceptance. Expanding AI applications to other areas of thyroid cancer research, such as risk stratification and the discovery of novel therapeutic targets, holds great potential for improving patient outcomes. AI will play an increasingly integral role in the investigation and management of MTC, transforming the landscape of thyroid cancer care”. |
Reviewer 2 Report
Comments and Suggestions for Authors
The authors reviewed holomics and AI-driven precision oncology for medullary thyroid carcinoma. The article gives a good overview of the cancer and the associated treatment, but the AI-driven precision oncology does not really reflect the current stage of their achievement, in particular on the AI-based diagnosis. Listed below are good examples that reflect the current AI-based diagnosis:
1. A pathology foundation model for cancer diagnosis and prognosis prediction
https://www.nature.com/articles/s41586-024-07894-z
2. Artificial intelligence for digital and computational pathology
https://www.nature.com/articles/s44222-023-00096-8?fromPaywallRec=false
3. An interpretable machine learning system for colorectal cancer diagnosis from pathology slides
https://www.nature.com/articles/s41698-024-00539-4?fromPaywallRec=false
Additionally, the figures are not properly made. For instance, the resolution of Figure 3 is too low. The input data for AI-driven diagnosis should be pathology slides or other samples taken from the patient. The input data should not be “Democracy,” although it is very important to have. Also, “Simultaneous Analysis” and “Computer-Aided Decision Making” appeared at very strange places of the graph. They should not be related to the input nor the output. To be honest, this type of figure only reflects that the authors lack of knowledge on AI-related topics. Perhaps focusing on the authors’ specialties and how AI-tools have been applied to address the difficulties in their work would be better?
Author Response
Reviewer 2
The authors reviewed holomics and AI-driven precision oncology for medullary thyroid carcinoma. The article gives a good overview of the cancer and the associated treatment, but the AI-driven precision oncology does not really reflect the current stage of their achievement, in particular on the AI-based diagnosis. Listed below are good examples that reflect the current AI-based diagnosis:
- A pathology foundation model for cancer diagnosis and prognosis prediction
https://www.nature.com/articles/s41586-024-07894-z
- Artificial intelligence for digital and computational pathology
https://www.nature.com/articles/s44222-023-00096-8?fromPaywallRec=false
- An interpretable machine learning system for colorectal cancer diagnosis from pathology slides
https://www.nature.com/articles/s41698-024-00539-4?fromPaywallRec=false
Additionally, the figures are not properly made. For instance, the resolution of Figure 3 is too low. The input data for AI-driven diagnosis should be pathology slides or other samples taken from the patient. The input data should not be “Democracy,” although it is very important to have. Also, “Simultaneous Analysis” and “Computer-Aided Decision Making” appeared at very strange places of the graph. They should not be related to the input nor the output. To be honest, this type of figure only reflects that the authors lack of knowledge on AI-related topics. Perhaps focusing on the authors’ specialties and how AI-tools have been applied to address the difficulties in their work would be better?
|
Reviewer 2 |
|
|
Recommendation |
Response |
|
1. authors reviewed holomics and AI-driven precision oncology for medullary thyroid carcinoma. The article gives a good overview of the cancer and the associated treatment, but the AI-driven precision oncology does not really reflect the current stage of their achievement, in particular on the AI-based diagnosis. Listed below are good examples that reflect the current AI-based diagnosis:
|
Noted. Thank you. We sincerely appreciate the inputs. |
|
2. A pathology foundation model for cancer diagnosis and prognosis prediction https://www.nature.com/articles/s41586-024-07894-z
|
Thank you. We checked it and found it valuable. |
|
3. An interpretable machine learning system for colorectal cancer diagnosis from pathology slides https://www.nature.com/articles/s41698-024-00539-4?fromPaywallRec=false
|
Thank you. We checked it and found it valuable. |
|
4. An interpretable machine learning system for colorectal cancer diagnosis from pathology slides https://www.nature.com/articles/s41698-024-00539-4?fromPaywallRec=false
|
Thank you. We checked it and found it valuable. |
|
5. Additionally, the figures are not properly made. For instance, the resolution of Figure 3 is too low. The input data for AI-driven diagnosis should be pathology slides or other samples taken from the patient. The input data should not be “Democracy,” although it is very important to have. Also, “Simultaneous Analysis” and “Computer-Aided Decision Making” appeared at very strange places of the graph. They should not be related to the input nor the output. To be honest, this type of figure only reflects that the authors lack of knowledge on AI-related topics. Perhaps focusing on the authors’ specialties and how AI-tools have been applied to address the difficulties in their work would be better?
|
Noted. Thank you. We accept the comment. We have deleted both Figure 1 and Figure 3. The information they contained is now comprehensively covered in the text. |
Reviewer 3 Report
Comments and Suggestions for Authors
The author describes the challenges faced by MTC patients during diagnostic evaluation and management, highlighting how hologenomics and artificial intelligence can help improve patient outcomes. By simultaneously analyzing and integrating results from biochemical, radiological, and histological studies, genetic research, and other sources along with individual patient information, AI can enhance the decision-making process. This innovative approach has the potential to personalize and optimize treatment strategies, leading to better management and outcomes for MTC patients. The thorough examination and creative method in the article were truly unforgettable. However, there are deficiencies in the structure and specifics of the article. Next, let me enumerate these problems:
1. The description of the data in Figure 1 in the introduction does not match the content of the figure; it is recommended to modify it.
2. The formatting of some abbreviations within parentheses is incorrect.
3. The introductory paragraphs under the main headings of Section 2 and Section 3 should generally outline the content described in the subsequent subsections.
4. There are minor errors in some text, such as missing punctuation or case errors. For example, in Section 2.1, "A low-risk sporadic MTC must be unifocal, low-grade tu and less than 1cm in maximum diA low-risk MTC must also be ameter. intra-thyroidal, and not harbour high-risk mutations," and in Section 2.2, "and re-do sur-gery 51,59-61];" as well as in Section 5, "… compression symptoms [137]. MTC. Using a …"
5. The chapter description in Section 5, including some of its content, does not align with the title; ultrasound examination and CT do not fall under the category of artificial intelligence mentioned in the title.
6. The font in Figure 3 is incorrect.
7. Abbreviations that have already been introduced in the text should be used directly in subsequent mentions.
8. It is recommended that the authors include a critical analysis of the limitations associated with the current AI models, especially in the context of data scarcity, which is a common challenge in rare diseases like MTC. This will provide a more balanced view and help set realistic expectations for the application of AI in this field. It is suggested to cite these literatures as follows.
[1] X. Fei, J. Wang, S. Ying, Z. Hu, J. Shi. Projective parameter transfer based sparse multiple empirical kernel learning Machine for diagnosis of brain disease. Neurocomputing. 413 (2020) 271-83.
[2] Bing Z, Brucker M, Morin F O, et al. Complex robotic manipulation via graph-based hindsight goal generation[J]. IEEE transactions on neural networks and learning systems, 2021, 33(12): 7863-7876.
[3] Lakhan A, Mohammed M A, Kozlov S, et al. Mobile‐fog‐cloud assisted deep reinforcement learning and blockchain‐enable IoMT system for healthcare workflows[J]. Transactions on Emerging Telecommunications Technologies, 2021: e4363.
[4] Fragkiadaki E, Anagnostopoulos F, Triliva S. The experience of psychological therapies for people with multiple sclerosis: A mixed‐methods study towards a patient‐centred approach to exploring processes of change[J]. Counselling and Psychotherapy Research, 2023.
9. "AI" is a particularly broad term, and the descriptions of various applications in this article mostly stop at the term "AI." Readers will not gain an understanding of the specific applications from this article. Additionally, several concepts that do not belong to AI, such as CT and ultrasound examination, are described as AI.
Author Response
Reviewer 3
The author describes the challenges faced by MTC patients during diagnostic evaluation and management, highlighting how hologenomics and artificial intelligence can help improve patient outcomes. By simultaneously analyzing and integrating results from biochemical, radiological, and histological studies, genetic research, and other sources along with individual patient information, AI can enhance the decision-making process. This innovative approach has the potential to personalize and optimize treatment strategies, leading to better management and outcomes for MTC patients. The thorough examination and creative method in the article were truly unforgettable. However, there are deficiencies in the structure and specifics of the article. Next, let me enumerate these problems:
- The description of the data in Figure 1 in the introduction does not match the content of the figure; it is recommended to modify it.
- The formatting of some abbreviations within parentheses is incorrect.
- The introductory paragraphs under the main headings of Section 2 and Section 3 should generally outline the content described in the subsequent subsections.
- There are minor errors in some text, such as missing punctuation or case errors. For example, in Section 2.1, "A low-risk sporadic MTC must be unifocal, low-grade tu and less than 1cm in maximum diA low-risk MTC must also be ameter. intra-thyroidal, and not harbour high-risk mutations," and in Section 2.2, "and re-do sur-gery 51,59-61];" as well as in Section 5, "… compression symptoms [137]. MTC. Using a …"
- The chapter description in Section 5, including some of its content, does not align with the title; ultrasound examination and CT do not fall under the category of artificial intelligence mentioned in the title.
- The font in Figure 3 is incorrect.
- Abbreviations that have already been introduced in the text should be used directly in subsequent mentions.
- It is recommended that the authors include a critical analysis of the limitations associated with the current AI models, especially in the context of data scarcity, which is a common challenge in rare diseases like MTC. This will provide a more balanced view and help set realistic expectations for the application of AI in this field. It is suggested to cite these literatures as follows.
[1] X. Fei, J. Wang, S. Ying, Z. Hu, J. Shi. Projective parameter transfer based sparse multiple empirical kernel learning Machine for diagnosis of brain disease. Neurocomputing. 413 (2020) 271-83.
[2] Bing Z, Brucker M, Morin F O, et al. Complex robotic manipulation via graph-based hindsight goal generation[J]. IEEE transactions on neural networks and learning systems, 2021, 33(12): 7863-7876.
[3] Lakhan A, Mohammed M A, Kozlov S, et al. Mobile‐fog‐cloud assisted deep reinforcement learning and blockchain‐enable IoMT system for healthcare workflows[J]. Transactions on Emerging Telecommunications Technologies, 2021: e4363.
[4] Fragkiadaki E, Anagnostopoulos F, Triliva S. The experience of psychological therapies for people with multiple sclerosis: A mixed‐methods study towards a patient‐centred approach to exploring processes of change[J]. Counselling and Psychotherapy Research, 2023.
- "AI" is a particularly broad term, and the descriptions of various applications in this article mostly stop at the term "AI." Readers will not gain an understanding of the specific applications from this article. Additionally, several concepts that do not belong to AI, such as CT and ultrasound examination, are described as AI.
|
Reviewer 3 |
|
|
Comment |
Response |
|
The author describes the challenges faced by MTC patients during diagnostic evaluation and management, highlighting how hologenomics and artificial intelligence can help improve patient outcomes. By simultaneously analyzing and integrating results from biochemical, radiological, and histological studies, genetic research, and other sources along with individual patient information, AI can enhance the decision-making process. This innovative approach has the potential to personalize and optimize treatment strategies, leading to better management and outcomes for MTC patients. The thorough examination and creative method in the article were truly unforgettable. However, there are deficiencies in the structure and specifics of the article. Next, let me enumerate these problems:
|
|
|
1. The description of the data in Figure 1 in the introduction does not match the content of the figure; it is recommended to modify it.
|
1. Thank you. We sincerely appreciate the comments. We have removed Figure 1. |
|
2. The formatting of some abbreviations within parentheses is incorrect.
|
2. Noted. We have re-checked and made corrections. |
|
3. The introductory paragraphs under the main headings of Section 2 and Section 3 should generally outline the content described in the subsequent subsections. |
3. Noted. Thank you. |
|
4. There are minor errors in some text, such as missing punctuation or case errors. For example, in Section 2.1, "A low-risk sporadic MTC must be unifocal, low-grade tu and less than 1cm in maximum diA low-risk MTC must also be ameter. intra-thyroidal, and not harbour high-risk mutations," and in Section 2.2, "and re-do sur-gery 51,59-61];" as well as in Section 5, "… compression symptoms [137]. MTC. Using a …"
|
4. Noted. Thank you. We have made corrections (marked in yellow). |
|
5. The chapter description in Section 5, including some of its content, does not align with the title; ultrasound examination and CT do not fall under the category of artificial intelligence mentioned in the title.
|
5. Noted. We however |
|
6. The font in Figure 3 is incorrect.
|
6. Noted and agreed. Thank you. Figure 3 has been removed. |
|
7. Abbreviations that have already been introduced in the text should be used directly in subsequent mentions.
|
7. Noted and agreed. Thank you. We have rechecked the ma |
|
8. It is recommended that the authors include a critical analysis of the limitations associated with the current AI models, especially in the context of data scarcity, which is a common challenge in rare diseases like MTC. This will provide a more balanced view and help set realistic expectations for the application of AI in this field. It is suggested to cite these literatures as follows.
[1] X. Fei, J. Wang, S. Ying, Z. Hu, J. Shi. Projective parameter transfer based sparse multiple empirical kernel learning Machine for diagnosis of brain disease. Neurocomputing. 413 (2020) 271-83. [2] Bing Z, Brucker M, Morin F O, et al. Complex robotic manipulation via graph-based hindsight goal generation[J]. IEEE transactions on neural networks and learning systems, 2021, 33(12): 7863-7876. [3] Lakhan A, Mohammed M A, Kozlov S, et al. Mobile‐fog‐cloud assisted deep reinforcement learning and blockchain‐enable IoMT system for healthcare workflows[J]. Transactions on Emerging Telecommunications Technologies, 2021: e4363. [4] Fragkiadaki E, Anagnostopoulos F, Triliva S. The experience of psychological therapies for people with multiple sclerosis: A mixed‐methods study towards a patient‐centred approach to exploring processes of change[J]. Counselling and Psychotherapy Research, 2023.
|
8. Noted. Thank you. We have covered some of the limitations of application of AI at the end of the discussion section. We also added a reference dealing specifically with the limitations of AI in medicine. We have read the four references suggested but did not add them to our bibliography. |
|
9. "AI" is a particularly broad term, and the descriptions of various applications in this article mostly stop at the term "AI." Readers will not gain an understanding of the specific applications from this article. Additionally, several concepts that do not belong to AI, such as CT and ultrasound examination, are described as AI.
|
9. Noted. Thank you. We have included examples of machine learning (AI) algorithms in the text and Table 1 and Table 3. |
Round 2
Reviewer 2 Report
Comments and Suggestions for Authors
Since the authors still ignore the current development on AI-driven cancer detection in their review, I have no more comments.
Author Response
The authors reviewed holomics and AI-driven precision oncology for medullary thyroid carcinoma. The article gives a good overview of the cancer and the associated treatment, but the AI-driven precision oncology does not really reflect the current stage of their achievement, in particular on the AI-based diagnosis. Listed below are good examples that reflect the current AI-based diagnosis:
- A pathology foundation model for cancer diagnosis and prognosis prediction
https://www.nature.com/articles/s41586-024-07894-z
- Artificial intelligence for digital and computational pathology
https://www.nature.com/articles/s44222-023-00096-8?fromPaywallRec=false
- An interpretable machine learning system for colorectal cancer diagnosis from pathology slides
https://www.nature.com/articles/s41698-024-00539-4?fromPaywallRec=false
Additionally, the figures are not properly made. For instance, the resolution of Figure 3 is too low. The input data for AI-driven diagnosis should be pathology slides or other samples taken from the patient. The input data should not be “Democracy,” although it is very important to have. Also, “Simultaneous Analysis” and “Computer-Aided Decision Making” appeared at very strange places of the graph. They should not be related to the input nor the output. To be honest, this type of figure only reflects that the authors lack of knowledge on AI-related topics. Perhaps focusing on the authors’ specialties and how AI-tools have been applied to address the difficulties in their work would be better?
Since the authors still ignore the current development on AI-driven cancer detection in their review, I have no more comments.
Response: We note the comment of the reviewer. It may not be completely correct that ‘we are ignoring current development on AI-driven cancer development’. What we indicated was that we did not deem it necessary to add the three articles that the reviewer suggested into our bibliography. We already had 52 references dealing specifically with AI, including coverage of digitalization of slides and whole slide emerging. Please check reference 101-170. Very few references are not covering AI related topics. We therefore think that references on AI are enough. We accept the opinion of the reviewer that our knowledge of AI may be limited.
Reviewer 3 Report
Comments and Suggestions for Authors
The manuscript presents an in-depth analysis of the application of holomics and artificial intelligence (AI) in the management of medullary thyroid carcinoma (MTC), a rare and aggressive form of thyroid cancer. The authors have successfully highlighted the importance of an integrated approach to improve patient outcomes in MTC. Below are my comments and suggestions for minor revisions:
1. The manuscript would benefit from a more detailed explanation of how AI algorithms are currently being used in the diagnostic workup of MTC. Could the authors provide more specific examples or case studies that demonstrate the impact of AI on diagnostic accuracy?
2. While the manuscript touches on the use of AI in healthcare, it would be valuable to include a discussion on the ethical considerations of using AI, particularly regarding data privacy and security in the context of MTC management. A section addressing these concerns would be appropriate.
3. The paper primarily focuses on the clinical and technological aspects of MTC management. Including patient perspectives on the use of AI and holomics in their treatment could provide a more holistic view of the impact. If possible, the authors should consider adding a section that discusses patient experiences or expectations.
4. The manuscript would be strengthened by citing more recent studies that have explored the integration of AI in cancer treatment, particularly those focusing on rare cancers like MTC. For instance, (i) Ying Chen, Taohui Zhou, Yi Chen, et al. HADCNet: Automatic Segmentation of COVID-19 Infection Based on a Hybrid Attention Dense Connected Network with Dilated Convolution. ii) X. Fei, J. Wang, S. Ying, Z. Hu, J. Shi. Projective parameter transfer based sparse multiple empirical kernel learning Machine for diagnosis of brain disease. Neurocomputing. 413 (2020) 271-83. (iii)Beibei Shi, Jingjing Chen, Yi Chen, et al. Prediction of recurrent spontaneous abortion using evolutionary machine learning with joint self-adaptive sime mould algorithm. The paper could benefit from referencing the work by these literatures on machine learning prediction models, which provides a contemporary perspective on AI's role in clinical diagnosis and treatment.
5. The figures and tables are informative but could be better integrated into the text. It is recommended that each figure and table be preceded by a brief introduction and followed by a discussion of its significance in the context of the paper.
6. In the results section, the authors mention various statistical tests used to analyze the data. It would be helpful to include a subsection on statistical methods that provides details on the rationale for choosing specific tests and the software used for the analysis.
7. The manuscript is generally well-written, but there are a few instances of awkward phrasing and typographical errors. I recommend a thorough proofreading and language editing to ensure clarity and readability.
8. The conclusion is succinct but could be expanded to summarize the key points made in the paper and reiterate the significance of the findings. It would also be beneficial to include a brief discussion on future research directions or areas where further investigation is needed.
Author Response
Reviewer 3
The manuscript presents an in-depth analysis of the application of holomics and artificial intelligence (AI) in the management of medullary thyroid carcinoma (MTC), a rare and aggressive form of thyroid cancer. The authors have successfully highlighted the importance of an integrated approach to improve patient outcomes in MTC. Below are my comments and suggestions for minor revisions:
- The manuscript would benefit from a more detailed explanation of how AI algorithms are currently being used in the diagnostic workup of MTC. Could the authors provide more specific examples or case studies that demonstrate the impact of AI on diagnostic accuracy?
Response: Thank you very much. We appreciate the inputs. We have expanded the manuscripts to include commonly used AI models, the processes of extracting predictive features and how performance of AI models is analyzed. We also included examples of how AI is used to address challenges of inconclusive fine needle aspiration cytology results, and generation of a nomogram for use to accurately predict the existence of lymph node metastasis.
- While the manuscript touches on the use of AI in healthcare, it would be valuable to include a discussion on the ethical considerations of using AI, particularly regarding data privacy and security in the context of MTC management. A section addressing these concerns would be appropriate.
Response: Noted. Thank you. Some of the ethical issues have been added in the subsection on limitations.
- The paper primarily focuses on the clinical and technological aspects of MTC management. Including patient perspectives on the use of AI and holomics in their treatment could provide a more holistic view of the impact. If possible, the authors should consider adding a section that discusses patient experiences or expectations.
Response: Thank you. We appreciate the suggestion. We searched for published work on patients experiences and expectations. Unfortunately, we only found one that we consider relevant. The paper covers doctors, nurses and patients experience with AI-assisted management of cardiac patients on anticoagulation. We have added it under the limitations.
- The manuscript would be strengthened by citing more recent studies that have explored the integration of AI in cancer treatment, particularly those focusing on rare cancers like MTC. For instance, (i) Ying Chen, Taohui Zhou, Yi Chen, et al. HADCNet: Automatic Segmentation of COVID-19 Infection Based on a Hybrid Attention Dense Connected Network with Dilated Convolution. ii) X. Fei, J. Wang, S. Ying, Z. Hu, J. Shi. Projective parameter transfer based sparse multiple empirical kernel learning Machine for diagnosis of brain disease. Neurocomputing. 413 (2020) 271-83. (iii)Beibei Shi, Jingjing Chen, Yi Chen, et al. Prediction of recurrent spontaneous abortion using evolutionary machine learning with joint self-adaptive sime mould algorithm. The paper could benefit from referencing the work by these literatures on machine learning prediction models, which provides a contemporary perspective on AI's role in clinical diagnosis and treatment.
Response: Thank you. We searched for the the articles and read them. We found them useful and have included them in the bibliography.
- The figures and tables are informative but could be better integrated into the text. It is recommended that each figure and table be preceded by a brief introduction and followed by a discussion of its significance in the context of the paper.
Response: Thank you. We sincerely appreciate the comment and recommendations. We have expanded the text before every table or figure.
- In the results section, the authors mention various statistical tests used to analyze the data. It would be helpful to include a subsection on statistical methods that provides details on the rationale for choosing specific tests and the software used for the analysis.
Response: Thank you. Noted. Our article is a review. The results included are from work done by some of the authors we cited.
- The manuscript is generally well-written, but there are a few instances of awkward phrasing and typographical errors. I recommend a thorough proofreading and language editing to ensure clarity and readability.
Response: Noted. Thank you. We have gone through the manuscript and indeed pick-up several typos, which we hope to have corrected.
- The conclusion is succinct but could be expanded to summarize the key points made in the paper and reiterate the significance of the findings. It would also be beneficial to include a brief discussion on future research directions or areas where further investigation is needed.
Response: Noted. Thank you. We have expanded the conclusion and included possible future directions.
NB: Please note that all changes we made are marked yellow.